# Dimethyl Fumarate as Potential Treatment for Alzheimer’s Disease: Rationale and Clinical Trial Design

**DOI:** 10.3390/biomedicines11051387

**Published:** 2023-05-08

**Authors:** Robert Sharkus, Richa Thakkar, Dennis L. Kolson, Cris S. Constantinescu

**Affiliations:** 1Department of Neurology, Cooper Neurological Institute, Cherry Hill, NJ 08002, USA; sharkus-robert@cooperhealth.edu (R.S.); thakkar-richa@cooperhealth.edu (R.T.); 2Department of Neurology, Perelman School of Medicine, University of Pennsylvania, Philadelphia, PA 19104, USA; dennis.kolson@pennmedicine.upenn.edu; 3Department of Neurology, Cooper Medical School of Rowan University, Camden, NJ 08103, USA

**Keywords:** Alzheimer’s Disease, microglia, NRF-2, heme oxygenase, dimethyl fumarate

## Abstract

Alzheimer’s Disease (AD) is a debilitating disease that leads to severe cognitive impairment and functional decline. The role of tau hyperphosphorylation and amyloid plaque deposition in the pathophysiology of AD has been well described; however, neuroinflammation and oxidative stress related to sustained microglial activation is thought to play a significant role in the disease process as well. NRF-2 has been identified in modulating the effects of inflammation and oxidative stress in AD. Activation of NRF-2 leads to an increased production of antioxidant enzymes, including heme oxygenase, which has been shown to have protective effects in neurodegenerative disorders such as AD. Dimethyl fumarate and diroximel fumarate (DMF) have been approved for the use in relapsing–remitting multiple sclerosis. Research indicates that they can modulate the effects of neuroinflammation and oxidative stress through the NRF-2 pathway, and as such, could serve as a potential therapeutic option in AD. We propose a clinical trial design that could be used to assess DMF as a treatment option for AD.

## 1. Overview of Alzheimer’s Disease 

Alzheimer’s Disease (AD) is a neurodegenerative disorder characterized by progressive cognitive impairment. It is the most common cause of dementia (accounting for 50–75% of cases), which affects 44 million people presently and is predicted to triple in prevalence by 2050. Annual healthcare costs related to AD were estimated to be USD 321 billion in the United States in 2022, with reasonable expectation for that cost to exponentially grow in the coming years [1]. In addition to the financial burden, AD can be extremely debilitating for patients, leading to loss of independence and functionality. It is a disease that not only affects the patient but can have a significant impact on the lives of their families and caregivers.

The etiology of AD is thought to be a combination of genetic and environmental risk factors. The hallmark pathophysiology of the disease is due to neuron loss; however, there have been many proposed mechanisms. Studies have shown that neuron loss results from a combination of amyloid deposition as well as neurofibrillary tangles from hyperphosphorylated tau protein. In AD, amyloid accumulates to higher levels than in normal aging and there is also amyloid deposition in the blood vessels [2]. More recently, neuroinflammation has been suggested to have a large role in the underlying pathology of AD. Microglia activation, stimulated and perpetuated by amyloid beta peptides and oligomers, mediates CNS neuroinflammation. While acute inflammation and microglia activation has been shown to have a protective effect against amyloid deposition, persistent microglial activation leads to an imbalance of pro- and anti-inflammatory cytokines which can ultimately worsen the disease process [3]. Additionally, oxidative stress related to inflammation plays a role in the early stage of AD, leading to amyloid deposition and tau hyperphosphorylation, further worsening the degeneration of neurons [4].

There are limited treatment options for AD. Currently, there are no medications which show a curative effect and many potential treatment options which show limited benefit or even harm. As outlined, this disease can be devastating to patients, families, and healthcare systems, and the need for more effective treatments is apparent. Recently, progress has been made and promising treatments are emerging [5,6]. These treatments target amyloid deposition through monoclonal antibodies that bind amyloid beta oligomers and aggregates. However, the antibody-bound aggregates and oligomers can deposit in blood vessels, leading to symptomatic or asymptomatic amyloid-related imaging abnormalities (ARIA) on magnetic resonance imaging (MRI) scans, characterized by edema or hemorrhage. There is evidence that these therapeutic antibodies, as well as spontaneously occurring anti-amyloid beta autoantibodies, lead to an increase in pro-inflammatory cytokines and microglia activation [7,8]. The now established role of neuroinflammation, microglial activation, and oxidative stress in the disease process offers an additional potential therapeutic target in the search for an effective treatment option [9].

## 2. Materials and Methods

A literature review was conducted to understand the pathophysiology of AD and highlight the importance of neuroinflammation and oxidative stress. Additionally, a search was conducted to evaluate the role of dimethyl fumarate (DMF), and by extension diroximel fumarate (DRF), on cognition overall and, more specifically, regarding AD pathophysiology. This search was completed utilizing the PubMed database ranging in dates from 1995 to 2023. Keywords were used to identify relevant studies, including combinations of “Alzheimer’s Disease”, “pathophysiology of Alzheimer’s Disease”, “microglial activation”, “oxidative stress”, “heme oxygenase”, “NRF-2”, “dimethyl fumarate”, “diroximel fumarate”, “cognition”, and “neuroinflammation”. Selected papers were reviewed by the authors to determine the relevance to the aim of the review and proposed clinical trial design.

## 3. Microglial Activation and AD

The most widely accepted mechanism of pathophysiology in AD is the deposition of amyloid beta (Aβ) plaques and neurofibrillary tangles caused by hyperphosphorylated tau. However, research has suggested that these processes alone do not explain the clinical manifestations and symptoms associated with AD [3,4]. Emerging data highlight the role of neuroinflammation and more specifically, microglial activation, as a key component to the disease process in AD. Of note, of the genes demonstrated through genome-wide association studies (GWAS) to be associated with susceptibility to AD, many are genes of the immune system and are expressed in microglia [10]. Microglia are the innate immune cells of the central nervous system (CNS) and are responsible for activation of the immune and inflammatory response in the brain. While typical microglia are only activated in the presence of a stimulus, studies have shown the microglia of aging brains can exhibit sustained activation [11]. Moreover, there appears to be a close relationship between microglial activation and tau hyperphosphorylation. This implies that microglia may not contribute to the disease process of AD through inflammatory processes alone but may also be linked to the formation of neurofibrillary tangles [12].

Although it was originally thought that microglia act in a dichotomized pro-inflammatory and anti-inflammatory state, the true function of these cells is far more complex. There is evidence showing a slightly protective effect of microglia in AD, in that they can target and act on amyloid plaques and phagocytose neurodegeneration debris [13]. However, microglial cells are often altered in the setting of AD, leading to morphologic changes and dysregulated behavior that leads to further damage and neurodegeneration. These dysregulated microglia, sometimes referred to as disease-associated microglia, can lead to the upregulation of certain genes, including triggering receptor expressed on myeloid cells 2 (TREM2), which is a crucial part of microglial activation [14]. Studies have shown conflicting results between TREM2 and the pathology in AD; however, their role in microglial activation suggests a link between the activation and progression of AD [15].

## 4. The Role of NRF-2

Sustained microglial activation leads to a pro-inflammatory state in the CNS, which ultimately results in the creation of reactive oxygen species (ROS) [16]. ROS are constantly being created in the body through natural metabolism. At times, they can be beneficial by aiding in the regulation of cell division, immune function, and inflammation. However, when ROS are produced in high amounts or without regulation, they can be harmful and cause cellular damage and death [17]. Prior studies have exhibited that lower levels of nitric oxide (NO) in the CNS exhibit a neuroprotective effect. Yet, in the presence of oxidative stress and ROS, NO can undergo a redox reaction forming reactive nitrogen species (RNS) which lead to neuronal damage and neurodegeneration [18]. In response to this oxidative stress, cells attempt to maintain homeostasis, which is regulated by a set of genes known as vitagenes. When oxidative stress from ROS is sensed, these vitagenes upregulate certain transcription factors, leading to the production of antioxidant and homeostatic enzymes [19,20]. Nuclear factor erythroid 2-related factor 2 (NRF-2) is a transcription factor that has been shown to regulate the antioxidant response. While NRF-2 is normally bound to a suppressor protein known as Kelch-like ECH associated protein 1 (KEAP1), increased oxidative stress leads to activation of NRF-2 through dissociation from the KEAP1 protein. Once unbound, NRF-2 then pairs with a deoxyribonucleic acid (DNA)-promoter binding sequence known as the human Antioxidant Response Element (ARE), which, in turn, activates several antioxidant enzymes, reducing the amount of ROS/RNS and limiting cellular damage and inflammation [21]. 

Oxidative stress has been described as a key component in many disease states, and more specifically, the reduction of oxidative stress through the NRF-2 pathway has been shown to exhibit protective effects in many diseases, including drug-induced liver injury, alcohol-induced liver disease, Parkinson’s Disease, and chronic obstructive pulmonary disease from cigarette smoking [17,22]. Additional studies have shown some effectiveness of natural supplements, such as phenylpropanoids, flavonoids, terpenoids, and alkaloids, in reducing endothelial cell injury through the NRF-2 pathway. This indicates that NRF-2 may play an important role in atherosclerosis and cardiovascular disease [23].

By reducing oxidative stress, the activation of NRF-2 can theoretically lead to decreased damage and neuronal loss in AD. However, while NRF-2 may play a role in the activation of TREM2, the effects of this activation remain unclear in AD. Regardless, the antioxidant and anti-inflammatory effects from enzyme activation through the NRF-2 pathway have the potential to counteract damage from sustained microglial activation [24].

## 5. The Role of Heme Oxygenase

One of the important pathways activated by NRF-2 is the heme oxygenase 1 (HO-1) pathway, which is outlined in Figure 1. Heme oxygenase exists in two isoforms: heme oxygenase-1 (HO-1), which is robustly inducible by multiple triggers that drive NRF-2 expression, and heme oxygenase-2 (HO-2), which is considered to be constitutively expressed and modestly inducible by few factors [25,26,27,28,29].

HO-1 and HO-2 are expressed in nearly all cell lineages, but only HO-1, the rapidly inducible isoform, is considered a critical mediator of the cellular response to injury [30]. Elevated HO-1 expression has been observed in brain tissue from individuals with AD, Parkinson’s disease, and multiple sclerosis (MS), perhaps reflecting a limited host protective response against ongoing injury [31,32,33]. The protective functions of HO-1 have been linked to the enzyme’s degradation of heme, a strong pro-oxidant, and the subsequent generation of the anti-inflammatory and anti-oxidative products, carbon monoxide, biliverdin, and bilirubin [30]. 

In the AD brain, HO-1 largely localizes to astrocytes, senile plaques, and neurofibrillary tangles. Enhanced astroglial expression of HO-1 has been described in patients with mild cognitive impairment (MCI). It directly correlates to decreased scores in global cognition and semantic and working memory, indicating its early pathogenesis in AD [34]. 

While the overexpression of HO-1 has been described in both MCI and AD, this may be a compensatory mechanism to neuronal loss in these disease states. Amyloid burden, pro-inflammatory cytokines, and malfunctioning of the electron transport chain in the mitochondria have been affiliated with the increase in oxidative stress in AD, leading to the upregulation of HO-1. This results in the degradation of heme, which has pro-oxidative properties, to its byproducts, which exert antioxidant effects [35,36]. In addition to the antioxidant effects, HO-1 can modulate neuroinflammation and has been associated with reduced tau expression and Aβ toxicity in rat models [37,38].

NRF-2 has been shown to directly control the expression of the HMOX1 gene which encodes HO-1. In vitro and in vivo studies have shown the significant role of NRF-2 in activating HO-1, which directly causes the anti-inflammatory effects of this pathway. Interventions that have shown upregulation of the NRF-2/HO-1 pathway have been neuroprotective in vitro and in animal models of AD. This suggests that the activation seen in AD is an attempt at activating endogenous antioxidant mechanisms [39].

## 6. Dimethyl Fumarate and Diroximel Fumarate

Dimethyl fumarate (DMF) was approved for treatment of relapsing–remitting multiple sclerosis (RRMS) in 2013. DMF is currently under study for the treatment of primary neurodegenerative diseases [40,41,42], inflammation-associated diseases [43,44,45,46], and for potential applications to SARS-CoV-2 infection [47,48,49,50,51,52]. DMF and its in vivo metabolite, monomethyl fumarate (MMF), suppress aerobic glycolysis by targeting glyceraldehyde 3-phosphate dehydrogenase (GAPDH) through irreversible succination of cysteine residues at the active site [53]. This DMF inhibition of GAPDH enzymatic function suppresses aerobic glycolysis in favor of anaerobic glycolysis to modulate immunity [53,54,55,56,57,58]. This has relevance for neuroprotection against disorders associated with disordered aerobic glycolysis and oxidative stress, including AD [59,60]. Moreover, cysteine succination of KEAP1, which anchors NRF-2 in the cytoplasm, releases NRF-2 for nuclear translocation, which results in transcriptional activation of NRF-2-inducible ARE-driven genes, including HO-1 [61].

The other immune-modulating effects of DMF and MMF include the induction of a Th1 to Th2 lymphocyte shift, inhibition of pro-inflammatory cytokine signaling, inhibition of nuclear factor kappa B (NF-κB) nuclear translocation, suppression of lymphocyte and endothelial adhesion molecule expression, and lymphocyte and monocyte chemotaxis [62,63]. They are also responsible for the induction of multiple NRF-2-dependent effector genes [40,41]. DMF and MMF mostly affect CD8+ T cells but are also known to affect CD4+ cells to a lesser degree, especially the pro-inflammatory T-helper Th1 and Th17 cells. The efficacy of DMF was studied in two randomized phase-III clinical trials: DEFINE and CONFIRM. It showed a reduction in annual relapse rates of RRMS by about 44 % (CONFIRM) and 53% (DEFINE). Both studies also showed a decrease in the number of MRI lesions while on DMF [64,65]. DMF specifically targets HO-1, which downregulates nitric oxide synthetase and reduces NFκB. Targeting HO-1 decreases the expression of IL-2 and stimulates T-regulatory cells. DMF was also found to be an agonist of hydroxycarboxylic acid receptor (HCAR), which reduces neuro-inflammation [66].

In addition to its immunomodulatory effects, DMF has been shown to have potent antioxidant effects [67]. In a recent study by Sun et al., DMF was studied as a proposed treatment option for AD in rodents. In mice treated to mimic AD, DMF was shown to increase neuronal survival after oxidative stress by several pathways. It reduced the production of ROS, decreased hippocampal atrophy, and inhibited the accumulation of Aβ deposition. The proposed mechanism of these results was facilitated through the NRF-2 pathway as the positive outcomes could not be reproduced in NRF-2 knockout mice. Additionally, NRF-2 wild-type mice treated with DMF had improved performance in the Morris water maze test over NRF-2 knockout mice, suggesting there could be a beneficial effect on processing speed and executive functioning. These results, as well as other research previously outlined, suggest that NRF-2 plays an important role in the pathogenesis of AD, and DMF could serve as a potential treatment option [68].

In both the DEFINE and CONFIRM trials, the most commonly reported side effects of DMF included flushing, nausea, diarrhea, and abdominal pain. Lymphocyte counts can be decreased with DMF use. However, regardless of the severe prolonged lymphopenia that was described, there was not a significant increased risk in serious infections [64,65]. Conversely, cases of progressive multifocal leukoencephalopathy (PML) have been reported in patients taking DMF [69].

Diroximel fumarate (DRF) is a novel oral fumarate that was approved for treatment of RRMS in 2019. Its slightly different chemical structure (as demonstrated in Figure 2) allows for the same active metabolite, MMF, but has less gut irritability and improved gastrointestinal (GI) tolerability. This was proven in the EVOLVE-MS-2 study, a phase-III randomized trial, which evaluated the GI tolerability of DRF when compared directly to DMF [70]. It is also known that in all stages of the disease, patients with MS suffer from cognitive impairment spanning several areas: memory, attention, information processing speeds, and executive function. However, there is evidence showing that MS patients treated with DMF can experience a slowing of cognitive impairment as well as improvement in quality of life and psychosocial function [71].

## 7. Rationale for a Clinical Trial of Dimethyl Fumarate/Diroximel Fumarate in Mild Cognitive Impairment and AD

The above features of DMF/DRF (from this point they will be collectively referred to as DMF in this article) make them appropriate candidates for clinical trials in AD.

Their neuroprotective, anti-inflammatory, and antioxidant effects of relevance to AD have been demonstrated through in vitro and in vivo studies. DMF has reduced ROS overproduction, reduced neuronal loss in NRF-2 knockdown neurons, and reduced amyloid beta-induced memory impairment and hippocampal atrophy, as seen in a murine AD model [64]. Similar results were obtained in other neurodegeneration models with cognitive impairment [72,73].

Garcia-Mesa et al. studied the neuroprotective effects of orally administered DMF in the non-human primate (Rhesus macaque) model of human immunodeficiency virus (HIV)-induced neurodegeneration. In this model, simian immunodeficiency virus (SIV), the primate homolog of HIV, causes simian acquired immunodeficiency syndrome (AIDS) and SIV-induced neurodegeneration. This is similar to the degeneration observed in HIV infection of humans. SIV-infected monkeys were administered oral DMF for up to 104 days, followed by necropsy and pathological analysis. This analysis showed significant reduction in oxidative stress, decreased expression of DNA markers, and downregulated protein oxidation in the frontal cortex of the treated animals. This was determined by optical redox imaging and immunohistochemical analyses of brain tissue in DMF-treated animals confirming a neuroprotective effect. This neuroprotective effect was observed in the absence of an obvious effect on microglia activation, as demonstrated by analysis of immune activation markers, CD68 and human leukocyte antigen DR isotype (HLA-DR); however, it is worth mentioning that CD8+ T cells were immunologically depleted in this model prior to SIV infection strictly to eliminate oxidative stress contributions from these cells. Thus, any CD8 cell-mediated microglia activation was suppressed in this model [67]. 

We propose taking DMF into clinical trials of early AD and its precursor stage, amnestic mild cognitive impairment (aMCI).

### 7.1. Phase I Trial

Although there is extensive experience with DMF drugs in MS, and a favorable safety profile has been well established, a Phase I trial of safety and tolerability is well justified given the differences between patient populations regarding age. The lymphopenia associated with DMF does not generally translate into a significant increase in risk of infections, including opportunistic infections. However, caution is needed in an older population.

A phase I trial design (outlined in Figure 3) would include randomizing patients to the standard dose of DMF used for MS vs placebo. Dosing for the medication arm would include 120 mg twice a day for 7 days followed by 240 mg twice a day for dimethyl fumarate, or 231 mg twice a day for 7 days followed by 462 mg twice a day for diroximel fumarate. The primary outcome measure would focus on safety and would require careful monitoring of lymphocyte counts (every 3 months and more frequently if there is lymphopenia with absolute lymphocyte count of less than 0.8) and monitoring of neurological function. Preliminary indicators of efficacy, used as a secondary outcome measure, would include standard cognitive measurements and MRI measurements of hippocampal volumes. 

Inclusion criteria would include early AD/aMCI (e.g., Mini Mental Status Exam (MMSE) ≥ 20). Exclusion criteria include the presence of other neurological disease (e.g., prior stroke, multiple sclerosis, etc.), psychiatric disease (e.g., schizophrenia, major depressive disorder), chronic infection, frequent recurrent infections (e.g., pneumonia), or history of malignancy, excluding non-melanoma skin cancer. The primary outcome measure would be safety, expressed as frequency of treatment-related severe adverse events, absolute lymphocyte count, immunoglobulins, and human polyomavirus 2 (JCV) antibody titers. Secondary outcome measures will be measures of brain volume using MRI (global volume, medial temporal lobe volume, and hippocampal volume), measured at baseline and 1 year, as well as a panel of cognitive tests, such as the Neuropsychological Test Battery (NTB) 5-item composite and the Alzheimer’s Disease Assessment Scale–Cognitive Subscale (ADAS-CS).

### 7.2. Phase II Trial

There are several ongoing phase II trials that target inflammation and microglia activation in AD. For example, AL002 is in a phase II trial (INVOKE-2) targeting microglial marker TREM-2, whose function is affected by a polymorphism associated with the risk of AD [74]. The design of a DMF phase II trial (outlined in Figure 4) would follow a similar design. 

Inclusion criteria would include (a) the diagnosis of early AD with evidence of brain amyloid by cerebrospinal fluid (CSF) analysis or positron emission tomography (PET) scan (if the serum amyloid beta 42/40 ratio (AD Detect-Quest Laboratories) is validated to replace CSF amyloid, it can be used instead/in addition); (b) an MMSE score ≥ 22 points or Clinical Dementia Rating (CDR) Global Score of 0.5–1.0; and (c) a study partner who consents to study participation and who cares for/visits the participant at least 10 h a week.

Written informed consent must be obtained and documented (from the participant or, where jurisdictions allow it, from their legal decision maker).

Exclusion criteria would include (a) dementia due to a condition other than AD including, but not limited to, frontotemporal dementia, Parkinson’s disease, dementia with Lewy bodies, Huntington disease, or vascular dementia; (b) other neurological disease (e.g., prior stroke, multiple sclerosis, etc.) or psychiatric disease (e.g., schizophrenia, major depressive disorder); (c) current uncontrolled hypertension, diabetes mellitus, or thyroid disease; (d) clinically significant heart disease, liver disease, or kidney disease; and (e) history of malignancy excluding non-melanoma skin cancer.

The primary outcome measure would be efficacy, measured by a comparison of functional scales at set time points (baseline, 1 year, and 3 years). The functional scales used would include the Alzheimer’s Disease Cooperative Study–Activities of Daily Living Scale (ADCS-ADL) and Clinical Dementia Rating Sum of Boxes scores (CDR-SOB). Secondary measures of efficacy would include measures using imaging, biomarkers, and cognitive assessments. Imaging measures would involve brain volume on MRI (at 1 year compared to baseline), translocator protein (TSPO) PET scan, and amyloid PET scan. Biomarkers would include plasma amyloid beta 42/40 ratio and plasma tau levels. Secondary cognitive assessment would include assessments such as the NTB and ADAS-CS, similar to the proposed phase I trial.

### 7.3. Future Considerations

As the therapeutic landscape in AD is likely to expand in the near future, in particular with agents targeting amyloid deposition, there will be increased need for treatments simultaneously targeting several key pathological features of AD, for example, amyloid accumulation and neuroinflammation, or to prevent the occurrence of ARIA, which also has an inflammatory component. We believe DMF would also be a good candidate as such an agent. A placebo-controlled trial in which all patients are on the amyloid-targeting antibody (e.g., lecanemab), with 1:1 randomization to DMF vs placebo, to determine the effect on the occurrence of ARIA as a primary measure and cognitive outcomes as secondary measures would be another potential avenue for exploration.

## 8. Conclusions

AD is an extremely taxing disease which has a significant impact on healthcare cost, quality of life, and independence for many people. The lack of effective treatment options and potential side effects of current treatments raises the question of considering a new target for therapy. As established through numerous studies and investigations, neuroinflammation, sustained microglial activation, and oxidative stress play a significant role in the pathophysiology of AD. As such, medications aimed at modulating these effects should be considered. There is sufficient evidence to suggest that activation of the NRF-2 pathway can potentially counteract the neurodegeneration seen in AD by attenuating these effects. A clinical trial investigating DMF as a treatment for patients with AD could offer another option for patients and families to improve quality of life in an otherwise devastating condition.

## Figures and Tables

**Figure 1 biomedicines-11-01387-f001:**
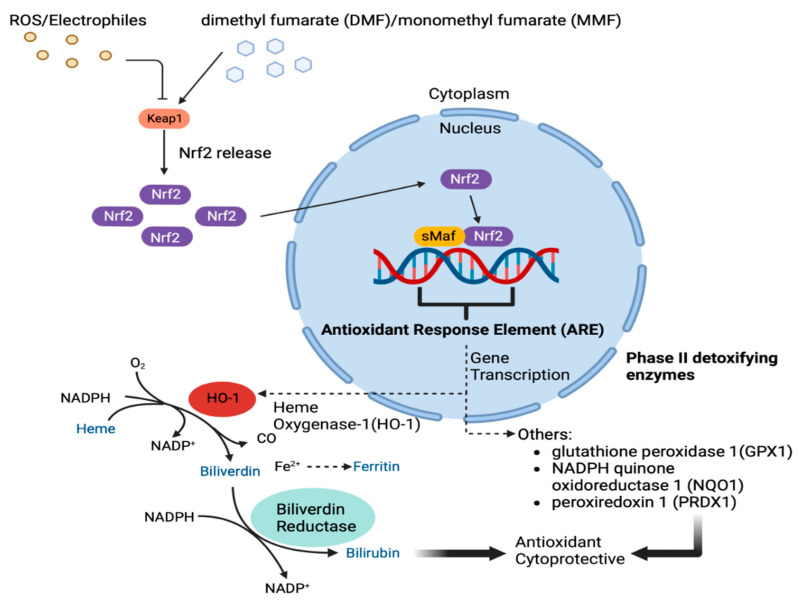
Cytoprotective pathways induced by dimethyl fumarate (DMF)/ monomethyl fumarate (MMF). Dimethyl fumarate (DMF) activity is mediated by its primary in vivo metabolite, monomethyl fumarate (MMF). Oral ingestion of DMF results in immediate demethylation within the small intestine to produce MMF, which is rapidly absorbed into the systemic circulation. After cellular uptake, MMF induces dissociation of KEAP1 from the inactive KEAP1/NRF-2 complex, thus releasing NRF-2 in the cytoplasm. Subsequent NRF-2 translocation to the nucleus results in binding to, and transcriptional activation of, the antioxidant response element (ARE) in the promoter region of numerous Phase II detoxifying enzymes. Among these, heme oxygenase 1 (HO-1) is a critical mediator of detoxification of heme, a major intra- and extra-cellular pro-oxidant produced during periods of metabolic stress. Heme cleavage results in the production of biliverdin, which is rapidly reduced to the diffusible antioxidant/cytoprotective product, biliverdin. Other phase II detoxifying enzymes (glutathione peroxidase 1/GPX1, NADP quinone oxidoreductase 1/NQO1, periredoxin 1/PRDX1) execute cytoprotective actions, including the reduction of peroxides and oxidant scavenging. Figure created with BioRender.com.

**Figure 2 biomedicines-11-01387-f002:**
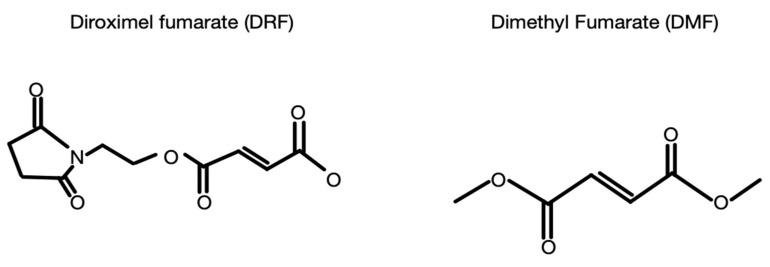
Schematic figure showing the chemical structures of dimethyl fumarate and diroximel fumarate. This difference in chemical structural allows for the same active metabolite with less side effects including GI irritability.

**Figure 3 biomedicines-11-01387-f003:**
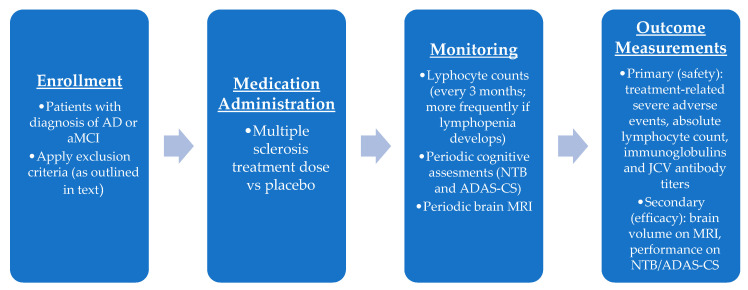
Outline of phase I clinical trial for DMF as a therapeutic option for AD/aMCI with primary focus on safety outcomes.

**Figure 4 biomedicines-11-01387-f004:**

Outline of phase II clinical trial for DMF as a therapeutic option for AD/aMCI with measurement of efficacy. Primary outcome would be functional status measured by ADCS-ADL and CDR-SOB, with secondary measures including imaging makers, plasma biomarkers, and cognitive assessments.

## Data Availability

No new data were generated in this work.

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
