# Peer review of "Dimethyl Fumarate as Potential Treatment for Alzheimer’s Disease: Rationale and Clinical Trial Design"

_biomedicines, 2023, doi:10.3390/biomedicines11051387_

Round 1
Reviewer 1 Report
In this work, Sharkus and colleagues reviewed the current literature regarding the emerging role played by DMF as a potential treatment for AD. This is a very interesting and well-written review proposing a clinical study design that could be used to evaluate DMF as a treatment option for AD.
Some minor revisions are needed:
-Typography: authors must carefully read their manuscript and check: 1) the space between words; 2) English of some sentences
- Please review all abbreviations in the text; often, some repeat and are not described the first time they appear (e.g. central nervous system lines 70-89; nrf2 line 119 etc.)
- Please add a graphic abstract to better explain the proposed clinical trial design.
- It would be useful to group the studies analyzing the role of DMF in AD in a summary table by dividing them into in vitro and in vivo studies, to make clearer the understanding of the rationale for the clinical trials.
In this work, Sharkus and colleagues reviewed the current literature regarding the emerging role played by DMF as a potential treatment for AD. This is a very interesting and well-written review proposing a clinical study design that could be used to evaluate DMF as a treatment option for AD.
Some minor revisions are needed:
-Typography: authors must carefully read their manuscript and check: 1) the space between words; 2) English of some sentences
- Please review all abbreviations in the text; often, some repeat and are not described the first time they appear (e.g. central nervous system lines 70-89; nrf2 line 119 etc.)
- Please add a graphic abstract to better explain the proposed clinical trial design.
- It would be useful to group the studies analyzing the role of DMF in AD in a summary table by dividing them into in vitro and in vivo studies, to make clearer the understanding of the rationale for the clinical trials.
Author Response
Response We thank the reviewer for these comments. Please find below the responses point by point:
Typography: authors must carefully read their manuscript and check: 1) the space between words; 2) English of some sentences
- Please review all abbreviations in the text; often, some repeat and are not described the first time they appear (e.g. central nervous system lines 70-89; nrf2 line 119 etc.)
- Please add a graphic abstract to better explain the proposed clinical trial design.
These changes have been made
- It would be useful to group the studies analyzing the role of DMF in AD in a summary table by dividing them into in vitro and in vivo studies, to make clearer the understanding of the rationale for the clinical trials.
Response: Due to the paucity of in vivo studies in experimental models relevant to AD, this has been done while retaining the narrative form
Reviewer 2 Report
In numerous experimental models, redox active compunds via hormetic dose responses display endpoints of biomedical and clinical relevance, which demonstrate to be endowed with powerful anti-inflammatory effects. Thus, interplay and coordination of redox interactions promoted by probiotics and their interaction with endogenous and exogenous antioxidant defence systems is an emerging area of reserach interest in anti-inflammatory anti-degenerative therapeutics. This reviewer is satisfied with the significance of this study, the care in which the study was performed, and the implications of the results for human health. Results presented are interesting and the questions posed are of extremely high interest, thus the paper does give adequate definitive information. Pending minor points, this paper can be accepted
Minor concerns:
1. Given the relationship between vitagene network and its possible biological relevance in the defense mechanisms against oxidative stress-driven degenerative diseases, Authors can mention in the discussion appropriately this aspect (See and quote please Calabrese V., et al., (2009) Biofactors 35, 146-160; ; Calabrese et al., 2010, Antiox. Redox Signal 13,1763; Calabrese et al., Nature Neurosci., 2007 8, 766; Siracusa R, et al., 2020, 9(9):824. doi: 10.3390/antiox9090824).
Author Response
- Given the relationship between vitagene network and its possible biological relevance in the defense mechanisms against oxidative stress-driven degenerative diseases, Authors can mention in the discussion appropriately this aspect (See and quote please Calabrese V., et al., (2009) Biofactors 35, 146-160; ; Calabrese et al., 2010, Antiox. Redox Signal 13,1763; Calabrese et al., Nature Neurosci., 2007 8, 766; Siracusa R, et al., 2020, 9(9):824. doi: 10.3390/antiox9090824).
Response: We thank the reviewer for the nice and insightful comments. The suggestions were followed and the references were included in the revised manuscript.
Reviewer 3 Report
Dear Authors,
clear, understandable, easy readable, thanks!
However, i\I would like to ask you to add some small improvements:
1) develop in special section/subsection (or put at the end of 1. section) the Plan with materials and methods described: manuscript Plan, search for the references (from-to), data basis used, inclusion/exclusion criteria for the papers, key words... This will guide the reader and will help better to understand the structure of the manuscript.
2) References - OK, just 3 "old ones", but they nicely fit into the context of the manuscript.
Author Response
1) develop in special section/subsection (or put at the end of 1. section) the Plan with materials and methods described: manuscript Plan, search for the references (from-to), data basis used, inclusion/exclusion criteria for the papers, key words... This will guide the reader and will help better to understand the structure of the manuscript.
2) References - OK, just 3 "old ones", but they nicely fit into the context of the manuscript.
Response: we thank the reviewer for the kind comments. We have included a materials and methods section which provides the literature search strategy.